# The Role of MicroRNAs in Breast Cancer and the Challenges of Their Clinical Application

**DOI:** 10.3390/diagnostics13193072

**Published:** 2023-09-28

**Authors:** Juan P. Muñoz, Pablo Pérez-Moreno, Yasmín Pérez, Gloria M. Calaf

**Affiliations:** 1Laboratorio de Bioquímica, Departamento de Química, Facultad de Ciencias, Universidad de Tarapacá, Arica 1000007, Chile; 2Programa de Comunicación Celular en Cáncer, Facultad de Medicina Clínica Alemana, Universidad del Desarrollo, Santiago 7780272, Chile; 3Instituto de Alta Investigación, Universidad de Tarapacá, Arica 1000000, Chile

**Keywords:** MicroRNAs, OncomiRs, MetastamiRs, breast cancer, diagnosis, cancer therapy

## Abstract

MicroRNAs (miRNAs) constitute a subclass of non-coding RNAs that exert substantial influence on gene-expression regulation. Their tightly controlled expression plays a pivotal role in various cellular processes, while their dysregulation has been implicated in numerous pathological conditions, including cancer. Among cancers affecting women, breast cancer (BC) is the most prevalent malignant tumor. Extensive investigations have demonstrated distinct expression patterns of miRNAs in normal and malignant breast cells. Consequently, these findings have prompted research efforts towards leveraging miRNAs as diagnostic tools and the development of therapeutic strategies. The aim of this review is to describe the role of miRNAs in BC. We discuss the identification of oncogenic, tumor suppressor and metastatic miRNAs among BC cells, and their impact on tumor progression. We describe the potential of miRNAs as diagnostic and prognostic biomarkers for BC, as well as their role as promising therapeutic targets. Finally, we evaluate the current use of artificial intelligence tools for miRNA analysis and the challenges faced by these new biomedical approaches in its clinical application. The insights presented in this review underscore the promising prospects of utilizing miRNAs as innovative diagnostic, prognostic, and therapeutic tools for the management of BC.

## 1. Introduction

In 1993, Victor Ambros and Gary Ruvkun discovered the first microRNA (miRNA), lin-4, in the nematode Caenorhabditis elegans, as a non-protein coding transcript that inhibited lin-14 gene expression without a reduction in mRNA levels [1,2]. However, the term miRNA was coined several years later to describe this class of small regulatory RNAs. It is now known that miRNAs are ubiquitously expressed in eukaryotes, and more than 2300 miRNAs have been identified in human cells, according to miRBase [3,4]. Notably, it is estimated that these miRNAs regulate approximately 60% of the human protein-coding genome [5].

Mature miRNAs are small RNA molecules, formed by about 19–25 nucleotides, involved in a wide range of biological processes, including differentiation, cell proliferation, apoptosis and stress response [6,7]. The canonical pathway by which miRNAs regulate gene expression is through a post-transcriptional mechanism, in which miRNAs bind to the 3’ untranslated region (UTR) of mRNAs and induce their degradation. This process is dominated by the eight-base seed region of the miRNA, which is the region of the miRNA that is most complementary to the target mRNA. When the miRNA binds to the 3’ UTR, it recruits proteins that degrade the mRNA [8,9]. In addition, alternative regulatory mechanisms have been described, such as the removal of polyadenine chains from the mRNA [10] and the regulation of translation initiation by binding to target sites in the coding region of mRNAs [11]. Thus, by establishing a complex miRNA–mRNA network, a single miRNA possesses the capability to directly or indirectly target numerous mRNAs, orchestrating several cellular processes at once [12]. In fact, It is suggested that approximately 100–200 target sites within the transcriptome can be recognized by each individual miRNA, and the inhibitory effect on gene expression can be achieved even with as few as 1000 copies of the miRNA present in a single cell [13].

The biogenesis of miRNAs is a well-studied process that has gained significant value for researchers in recent years. miRNAs can be produced through either the canonical pathway or one of several non-canonical pathways. In the canonical pathway, miRNAs are transcribed from miRNA-specific genes to produce long-stranded precursors called primary miRNAs (pri-miRNAs), which contain a 5’-cap and a poly-A tail [14,15]. The Microprocessor complex, composed of a single DROSHA and two DGCR8 molecules, is responsible for the cleavage and maturation of primary miRNAs (pri-miRNAs) into precursor miRNAs (pre-miRNAs). Within this process, DROSHA functions as a measuring gauge, precisely assessing an 11-base pair distance from the basal ssRNA–dsRNA junction [16,17]. Then, pre-miRNAs are transported to the cytoplasm, where they undergo further maturation by the RISC loading complex (RLC) [18]. During this process, one strand of the pre-miRNA duplex is preferentially selected as the mature miRNA, while the other strand is typically degraded [19]. The mature miRNA within the RISC complex guides it to specific messenger RNA (mRNA) molecules through base-pairing interactions [8].

In healthy cells, miRNAs are tightly regulated to maintain homeostasis [20]. However, miRNA dysregulation is a common feature of many diseases, including neurodegenerative diseases, metabolic diseases, and cancer [21,22,23,24,25,26,27]. In fact, the expression patterns of miRNAs can vary greatly between normal and cancerous tissues, as well as between localized and aggressive forms of cancer, and depending on the type and stage of the disease [28]. Interestingly, certain miRNAs have been shown to possess the ability to induce oncogenesis, while others are involved in regulating gene targets associated with metastasis [29]. Conversely, some miRNAs function as inhibitors of tumor growth, referred to as tumor suppressor miRNAs [30]. This suggests that miRNAs play a role in cancer progression and may be potential targets for therapeutic intervention [31].

Breast cancer (BC) is the most prevalent cancer in women worldwide [32]. Although numerous chemotherapeutic agents have been developed to combat BC, the primary limitation of these approaches is the development of resistance to these compounds and radiotherapy [33]. This poses a significant challenge, as it can make it difficult to effectively treat the disease. Consequently, there is a growing body of research aimed at addressing this challenge. One promising area of research is the development of targeted therapies that specifically target the molecular pathways that drive cancer growth [34,35]. These therapies have the potential to be more effective than traditional chemotherapeutic agents, and they may also be less likely to cause resistance [36].

In recent years, deep sequencing and profiling studies of miRNAs have revealed substantial evidence of miRNA dysregulation in BC [37,38]. This dysregulation is associated with a number of cancer hallmarks, including the evasion of growth suppressors, resistance to cell death, sustained proliferative signaling, activation of invasion, and induction of angiogenesis [39,40]. In addition, the levels of circulating miRNAs are recognized to revert to their initial levels after the removal of a tumor. This observation supports the potential value of circulating miRNAs as biomarkers for predicting prognosis in early stage BC and to assess the effectiveness of cancer treatments [41]. Therefore, knowing which miRNAs are overexpressed in different types of BC, as well as identifying the role of each of them in the processes of carcinogenesis, will allow their use as diagnostic and therapeutic tools.

## 2. miRNAs in BC Subtypes

BC is a major public health problem worldwide, with an estimated 2.3 million new cases and 685,000 deaths each year. Currently, it is the leading cause of cancer death in women [32]. From a clinical perspective, BC is a heterogeneous and multifactorial disease, characterized by a broad intertumoral and intertumoral nonuniformity and influenced by a variety of risk factors [42,43]. The identification of diagnostic biomarkers for early-stage detection, as well as the development of specific and sensitive therapeutic options, remain important challenges in this malignancy.

BC typically arises from the epithelial cells that line the milk ducts or lobules responsible for milk production [42]. As the disease progresses, the cancer cells can infiltrate neighboring tissues and disseminate to distant sites in the body via the lymphatic system or bloodstream [42]. BC encompasses different types with distinct gene expression patterns, behaviors, prognoses, and responses to therapy. BC classification is typically based on the expression of the estrogen receptor (ER), progesterone receptor (PR), and the human epidermal growth factor receptor 2 (HER2). This is known as histological stratification, and it has been a useful tool for historically guiding the treatment [44]. Thus, BCs can be classified into different subtypes based on the expression of ER, PR and HER2. These subtypes include ER+/PR+, HER2-positive and triple-negative BC (TNBC), which lacks the expression of all three receptors. In addition, BC subtypes are identified through transcriptome profiling, which has revealed four main subtypes: luminal A, luminal B, HER2-positive, and basal-like. The majority of ER+/PR+ BCs belong to the luminal A or luminal B subtypes, while TNBCs are mainly basal-like [45,46,47]. However, recent emerging findings from research on intertumoral heterogeneity suggest the potential presence of multiple co-existing BC subtypes within a single tumor [48].

Several studies have shown that the different subtypes of BC show a particular pattern of expression of miRNAs [49,50] (Table 1). These findings suggest that miRNAs could be used as biomarkers for BC subtypes and as targets for therapeutic intervention. However, it is important to note that miRNA expression profiles can vary within subtypes due to tumor heterogeneity and individual patient characteristics. 

## 3. Oncogenic, Tumor Suppressor and Metastatic miRNAs in BC Cells

miRNA dysregulation in cancer cells was first suggested by Calin et al., (2002) [65], who demonstrated that two miRNAs, miR-15a and miR-16-1, were located in a region commonly deleted in B-cell chronic lymphocytic leukemia patients [65]. Kumar et al., (2007) [66] later found that inhibiting miRNA maturation can promote cellular transformation and tumorigenesis, supporting the idea that miRNAs play a crucial role in cancer progression [66]. Currently, specific miRNA signatures have been associated with poor prognosis in several types of cancer, suggesting their potential as diagnostic and prognostic markers [67].

Certain miRNAs have the ability to function as oncogenes, actively contributing to the progression of cancer. These miRNAs are commonly referred to as “oncomiRs” [68] The role of oncomiRs involves various mechanisms that contribute to tumor progression. Firstly, they exert their influence by employing a negative inhibitory mechanism on tumor suppressor genes, thereby facilitating tumor development. Additionally, oncomiRs play a crucial role in governing the temporal aspects of cell differentiation, proliferation, and the cell cycle. Moreover, they regulate the expression of oncogenes [69] (Figure 1). 

There are a number of oncomiRs that have been identified in BC cells, including miR-9 [68], miR-10b [70], Cluster 17/20 [71], miR-21 [72], miR-155 [73], miR-221/222 [74], miR-210 [75] and miR-183 [76]. These miRNAs are thought to promote BC development by targeting tumor suppressor genes, which are genes that normally help to control cell growth and division [77]. For example, miR-10 has been shown to target the tumor suppressor gene HOXD10. HOXD10 is involved in cell–cell adhesion, which is key for maintaining the integrity of tissues. When miR-10 targets HOXD10, it disrupts cell–cell adhesion, which allows cancer cells to invade and spread [78]. 

In addition to their direct effects on tumor cells, oncomiRs can also influence the tumor microenvironment [79]. For example, miR-21 has been shown to promote tumor-associated angiogenesis by targeting the expression of anti-angiogenic factors, such as PTEN, thereby enhancing HIF-1α and VEGF expression [80]. This highlights the multifaceted nature of miRNA-mediated oncogenic effects in BC. 

On the other hand, some miRNAs can inhibit the expression of oncogenes that promote breast tumorigenesis. These miRNAs, known as tumor suppressor miRNAs, are involved in regulating various pathways related to cell proliferation, migration, invasion, and apoptosis [81] (Figure 1). For example, miR-203 has been shown to inhibit the expression of oncogenes such as BIRC5 and LASP1 in TNBC cell lines [82]. The downregulation of these miRNAs has been associated with poor prognosis and increased metastasis in BC patients [83]. 

In the past, researchers believed that miRNAs could be classified as either tumor suppressors or oncogenes based on their functions. However, recent studies have revealed a more complex picture, indicating that certain miRNAs can exhibit dual roles. A notable example of this dual functionality is miR-125b. Initially, miR-125b was recognized for its ability to inhibit cancer cell growth, as demonstrated in some research studies [84,85]. Yet, a more recent investigation has unveiled an intriguing twist, showing that miR-125b can also stimulate the growth of cancer cells in specific breast cancer (BC) cell contexts [86]. This shows that the function of miRNAs can be context-dependent. This makes it important to carefully consider the role of a miRNA in cancer before using it as a therapeutic target.

In addition, a growing body of research has identified a specific group of miRNAs that play a vital role in the initiation of metastasis, termed “metastamiRs” [87,88]. MetastamiRs have been recognized as crucial regulators of BC metastatic programming, as they exert control over genes that directly affect the ability of cancer cells to spread [89]. In addition, metastamiRs influence the expression of genes associated with EMT, a fundamental mechanism that endows epithelial cells with the attributes necessary for metastasis, such as motility and invasiveness. MetastamiRs have been shown to influence microenvironmental factors, which can either facilitate or hinder the survival and growth of migrating cancer cells. Lastly, these miRNAs are implicated in the secretion of exosomes, small vesicles that can transfer information and materials between cells and have a role in promoting metastasis.

In summary, miRNAs can play a dual role in BC: some can act as oncogenes, promoting tumor growth, invasion, metastasis, and therapeutic resistance, while others can also act as tumor suppressors, suppressing tumor growth and progression [77]. While some aberrant miRNAs may be shared across different types of cancer, cancer-specific miRNAs are the most common. Consequently, these types of miRNAs offer promising prospects for their utilization as therapeutic tools and potential indicators for diagnosis and prognosis in BC [90].

## 4. miRNAs and Its Potential Therapeutic Use in BC 

Conventional chemotherapy, while effective in inhibiting cancer cell growth, is limited by its non-specificity and associated side effects on healthy tissues. To address this, targeted therapy has been developed to specifically target molecular factors involved in tumor formation and progression. On the other hand, miRNAs offer a distinct advantage as they can efficiently silence target genes and regulate multiple genes of interest simultaneously. This capability is particularly beneficial in the treatment of cancer, which is characterized by its heterogeneity and diverse genetic components.

Since miRNAs can act as oncomiRs, tumor suppressors or metastamiRs, novel strategies have emerged to use them as potential therapeutic targets for the treatment of BC. There are two possible strategies for using miRNAs in therapeutic approaches.

### 4.1. Inhibiting OncomiRs and MetastamiRs

Through oncomiR and metastamiR inhibition, it is possible to restore the expression of tumor suppressor genes and inhibit oncogenic signaling pathways, thereby suppressing tumor growth and metastasis in BC cells [91,92,93,94]. For this purpose, different approaches have been explored, including 

Antisense oligonucleotide-targeting miRNAs (AMOs): AMOs are synthetic molecules designed to specifically inhibit the activity of miRNAs. They function by binding to the target miRNA through complementary base pairing and preventing its interaction with its target mRNA. By doing so, AMOs effectively block the function of the miRNA, modulating gene expression and cellular processes regulated by that miRNA. AMOs are typically chemically modified oligonucleotides that exhibit high specificity and stability.The use of small molecules: Small molecules can interact with proteins involved in miRNA biogenesis or bind to miRNA-specific secondary structures. They can be designed with the aid of bioinformatics tools or identified through the experimental screening of pharmacologically active chemical compounds [95].miRNA-based approaches with conventional drugs: Combining miRNA-based approaches with conventional drugs can improve the efficiency of drug-based therapies by targeting cellular pathways that affect therapeutic outcomes [96].AntagomiRs and siRNAs: The utilization of Antagomirs in conjunction with siRNAs presents an alternative approach to enhance the efficacy of therapeutic miRNAs [97]. Antagomirs are chemically modified RNA molecules that are complementary to specific miRNAs. They function by specifically binding to endogenous miRNAs and inhibiting their activity. siRNAs are a group of small RNAs that can specifically regulate a single or few target genes.miRNA sponges: miRNA sponges are RNA molecules that contain multiple binding sites that are complementary to specific miRNAs, effectively sequestering and titrating the miRNAs away from their natural targets. By doing so, miRNA sponges can indirectly regulate gene expression by preventing the binding of miRNAs to their target mRNAs [98].

Currently, several pre-clinical studies have demonstrated that inhibiting specific miRNAs can increase the sensitivity of BC cells to chemotherapy. For example, in MCF-7/ADR cell lines, the transfection of miR-3609 increased the susceptibility of tumor cells to adriamycin-based chemotherapy [99]. Similarly, in a study conducted by Lin et al. (2021), they achieved significant enhancement in chemotherapy sensitivity among 65 BC patients. This improvement was accomplished by inducing miR-133 expression in cisplatin-resistant TNBC cells obtained from these patients [100]. Thus, targeting oncomiRs presents a promising therapeutic approach to increase the sensitivity of BC cells to chemotherapy (Figure 2).

### 4.2. Restoring the Function of Tumor Suppressor miRNAs

The second strategy aims to restore the impaired function of tumor suppressor miRNAs by introducing miRNA mimetics into the tumor microenvironment. This approach is designed to suppress tumor growth and halt the spread of malignancy. Promising results have been demonstrated in both in vitro and animal studies, showing the effectiveness of this therapy. For example, Trepel et al., (2015) [101] developed a new type of tumor-specific adeno-associated viral vector (AAV) to deliver miRT-1d to cancer cells with greater precision. This “suicide gene” significantly inhibited tumor growth after one single vector administration, and it was more effective than using untargeted vectors and did not cause any side effects [101]. Likewise, Ramchandani et al., (2019) [102] investigated the use of multilayered nanoparticles (NPs) carrying miRNA miR-708 (miR708-NP) as a therapeutic approach for TNBC. They found that these NPs efficiently delivered miR-708 to reduce metastasis. These findings suggest an antimetastatic therapeutic approach using miR-708-NP, for high-risk TNBC patients in clinical settings [102].

Despite the remarkable progress made in the preclinical development of miRNA therapeutics in recent decades, their successful clinical application faces significant challenges. The main difficulties revolve around ensuring stability and effective delivery. This necessitates the exploration and development of new delivery strategies. The penetration of miRNAs into tumors has proven to be considerably inefficient. This is primarily due to the tumor’s permeable structure, which leads to inadequate blood perfusion. Additionally, miRNAs are inherently unstable and prone to degradation by nucleases in the bloodstream upon administration into the body. Moreover, tumor cells exhibit a low uptake of miRNAs, and there is the potential for unwanted effects from miRNA delivery. All of these factors create substantial barriers to the effective delivery of miRNAs for therapeutic purposes. To overcome these challenges, there are a number of different delivery strategies being explored, including viral vectors, nanoparticles, and liposomes. However, the best delivery strategy for a particular miRNA will depend on a number of factors, including the target cancer, the stability of the miRNA, and the desired therapeutic effect.

Although miRNA-based therapies hold promise for improving BC treatment outcomes, further investigations are required to gain a comprehensive understanding of the molecular pathways that can be modulated by miRNAs. These studies will contribute to a more comprehensive overview of miRNA-related mechanisms and their potential therapeutic applications in BC and beyond (Figure 2).

## 5. miRNAs as Diagnostic Markers of BC

The early diagnosis of BC is one of the most effective ways to prevent death from the disease. Mammography is the established method for early detection, and multiple randomized controlled trials have shown that mammographic screening significantly reduces BC mortality [103], but it has limitations such as false negatives, false positives, and unnecessary biopsies [104]. Breast ultrasound is used as an auxiliary test, but it relies on equipment, subjective interpretation and may lead to false positives [105]. Additionally, invasive biopsies have drawbacks, including patient discomfort and potential side effects [106]. Liquid biopsy has surfaced as a viable solution to address these limitations. It allows for noninvasive sample collection and enables early cancer detection and comprehensive monitoring [18,19]. Consequently, it offers simplicity, is painless and avoids the recovery periods and side effects of tissue biopsy [107,108,109,110]. In this context, circulating miRNAs have emerged as biomarkers for the early detection and diagnosis of BC in liquid biopsies [111,112,113,114]. Circulating miRNAs refer to extracellular miRNAs found in body fluids, including blood, serum, plasma, breast milk, saliva, and urine [115]. They exist either as free-circulating miRNAs, enclosed within extracellular vesicles (EVs) like exosomes or bound to a protein [116,117,118]. 

Cancer patients have been observed to have a higher concentration of tumor-derived exosomes circulating in their bloodstream compared to healthy individuals [119]. These exosomes contain miRNAs that have the potential to serve as circulating biomarkers for the early detection of various diseases [120,121]. In contrast to free miRNAs found in whole blood or serum, miRNAs enclosed within exosomes are more stable and dependable. The protective phospholipid bilayer surrounding exosomes shields them from degradation by nucleases in bodily fluids [122]. As a result, exosomal miRNAs have emerged as a promising biomarker for diagnosing BC, garnering increasing attention in research [123].

Several studies have shown that circulating miRNAs can be used as a minimally invasive tool for BC diagnosis (Table 2). Additionally, some studies have also shown that circulating miRNAs can be used to categorize the stage of BC [124,125,126] and predict the best treatment option [61,111,112,113,114,127,128]. In fact, some clinical trials are currently underway to evaluate circulating miRNAs as markers for the liquid biopsy of human BC. These trials are expected to provide further evidence of the potential of circulating miRNAs as a diagnostic and prognostic biomarker for BC. (https://www.clinicaltrials.gov/ accessed on 18 July 2023).

The role of miRNAs in the regulation of gene expression and their potential involvement in tumorigenesis provide a compelling avenue for investigating the pathophysiology of BC at a molecular level. This could reveal previously unidentified mechanisms of disease development and progression, thus opening the door to novel therapeutic targets. It is also noteworthy to mention the potential of these miRNAs in the monitoring of treatment responses. Given their biological stability in body fluids and changes in expression levels corresponding to disease progression or regression, they could serve as real-time, non-invasive biomarkers for monitoring treatment efficacy. This would allow for timely modifications of treatment plans, thereby improving patient outcomes. 

In summary, these findings underscore the potential utility of both free and EV miRNAs as groundbreaking tools in the early detection of BC. Notably, these miRNAs may assist in differentiating between various subtypes of the disease, thereby fostering personalized treatment strategies. Further, the unique expression patterns of these miRNAs could act as reliable ‘biomolecular signatures’ for individual BC subtypes [129]. Consequently, they could serve as a foundation for developing precise diagnostic assays which, combined with existing diagnostic techniques, might significantly enhance the sensitivity and specificity of BC detection. 

**Table 2 diagnostics-13-03072-t002:** Studies of circulating miRNAs as diagnostic biomarkers of BC.

Number of Samples (BC/HC)	Source	miRNAs	Results AUC Value/Sensitivity (%)/Specificity (%)	Ref.
226/146	Plasma	miR-373, miR-24, miR-206, and miR-1246	0.992/98/96	[127]
78/72	Serum	miR-21-5p, miR-23a-3p	0.92/79.5/100	[128]
755/86	Tissue	Panel of 28 miRNAs	NS/97–76/98–80	[130]
183/106	Serum	Panel of 30 miRNAs	0.915/72.2/91.5	[131]
177/197				
180/199				
68/13	Plasma	miR-185-5p, miR-362-5p	0.957/92.65/92.31	[132]
257/257	Plasma	miR-122-5p, miR-146b-5p, miR-210-3p, miR-215-5p, Let-7b-5p	0.843/81.1/78.4	[133]
102/53	Serum	miR-214	0.924/NS/NS	[134]
112/59	Serum	miR-103-3p	0.697/78.2/74.7	[135]
54/89	Plasma	miR-30b5p	0.77/57.4/87.54	[136]
		miR-99a5p		
33/37	Plasma	miR-1246	0.982/97.30/93.94	[137]
50/30	Serum	miR-106a	0.947/100/83.33	[138]
61/48	Serum	miR-10b	0.98/92.9/97.9	[139]
196/49	Serum	miRNA-373	0.98/90.8/98.4	[140]
40/40	Serum	miR-660-5p	0.774/79/61	[141]
		miR-210-3p	0.716/68/51	
35/33	Plasma	miR-145-5, miR-191-5p	0.984/94/100	[142]
102/15	Plasma	miR-155	NS	[143]

Abbreviations: AUC, area under the ROC curve; ROC curve, receiver-operating characteristic curve. HC, healthy control. NS, not specified.

## 6. Use of miRNAs as Prognostic Marker of BC Recurrence

BC recurrence is the reappearance of cancerous cells in or around the breast (local recurrence) or in distant parts of the body after the completion of the initial treatment. It both continues to be a major concern for BC patients and represents the principal cause of BC-related deaths [144]. Despite notable progress in BC treatment, a considerable number of patients still confront the daunting prospect of disease recurrence [145]. This highlights the crucial importance of identifying reliable biomarkers capable of accurately predicting recurrence risk and guiding personalized treatment strategies.

A recent study proposed that miRNAs might be fundamental elements involve in the cancer cell dormancy and latent metastases, which are related to the recurrence of BC [146]. As a result, knowing differentially expressed circulating miRNAs and tumor-based miRNAs in early-stage BC patients may offer predictive indicators of relapse before the manifestation of recurrence symptoms [147]. These miRNAs can influence abnormal cellular behavior, relapse risk, and, ultimately, survival outcomes. By conducting a comprehensive analysis of the interplay between miRNA levels, clinical conditions, and treatment responses, clinicians have the potential to predict BC patient survival and the likelihood of disease recurrence more effectively [148,149,150,151,152,153]. This proactive approach can significantly improve patient outcomes and reduce the burden of recurrent BC. 

Numerous studies have provided compelling evidence that the distinct expression of circulating and tumor miRNAs in patients with early-stage breast cancer has the ability to predict BC relapse even before the manifestation of clinical symptoms. This predictive ability derives from the ability of these miRNAs to exert significant influence on cancer cell behavior, vulnerability to relapse, and patient survival (Table 3). Although these preliminary findings suggest a significant role of miRNAs as potential biomarkers, further validation is still needed to confirm these results. Hence, future translational research should prioritize comprehensive assessments of miRNAs. 

## 7. Challenges for Therapeutic Use of miRNAs in BC 

Despite significant advancements in understanding the molecular biology of BC, the role of miRNAs in this particular neoplasm is still in its early stages. The inconsistent results observed in screening methods underline the need for optimized techniques to yield more accurate and informative outcomes in the future. For example, the evidence concerning the levels of miR-145 in BC is conflicting. Some studies show an increase in its expression in BC patients [142,162], while others indicate a decrease [114]. Thus, the lack of standardized and robust methodologies incorporating universal parameters to detect tumor-specific circulating miRNAs in body fluids could contribute to variations in the results of different studies. Therefore, larger and more comprehensive studies are still required to establish whether miRNAs can serve as noninvasive diagnostic biomarkers and predictors of therapeutic response in cancer patients.

Another significant challenge in utilizing miRNAs as biomarkers lies in their typically low abundance in blood, which also present intricate matrices with potential interfering biomolecules, like proteins. The process of distinguishing miRNAs from other types of RNAs further adds to the complexity [163]. In this regard, the use of a single miRNA for BC diagnosis leads to low sensitivity and specificity, leading to a reduced accuracy. A better approach is to combine multiple miRNAs, as some studies have demonstrated [127].

On the other hand, miRNA levels can be influenced by several factors, including age, gender, and the presence of disease, further complicating their use as reliable markers [164]. Another concern arises from the fact that certain potential miRNA biomarkers are ubiquitously present in both healthy individuals and those with cancer [165]. These ubiquitous miRNAs often display only minor variations in expression levels between healthy and diseased states. Thus, the methodology for sampling becomes critical in accurately distinguishing between health, malignant disease, and benign conditions. 

The complexities mentioned can result in inconsistencies when comparing findings across various studies, thereby posing additional hurdles in the standardization and broader adoption of miRNA-based diagnostics. In this context, several strategies are used to overcome this challenge, including employing an enrichment step prior to global expression profiling [166]. This process of increasing the concentration of miRNAs in a sample can improve the sensitivity of miRNA detection methods [163]. Another approach is to use next-generation sequencing (NGS). NGS allows for the identification of a wider range of miRNAs, including novel ones, at high sensitivity. However, NGS is more expensive and time-consuming than other methods. As an alternative, mass spectrometry (MS)-based techniques provide the multiplexed, direct detection and quantification of miRNAs. MS-based methods are superior in quantification, as it allows miRNA sequencing, quantitation, multiplexed detection, and the evaluation of post-transcriptional modifications [163]. 

On another front, the development of miRNA-based therapeutics, including miRNA mimics or inhibitors, presents significant potential for personalized treatment approaches [69]. These therapeutic interventions have the capacity to selectively target oncomiRs to suppress their activity or restore the function of tumor-suppressive miRNAs, thereby modulating critical cancer-related pathways. Furthermore, integrating miRNA profiling with existing genomic profiling platforms has the potential to enhance the accuracy of BC subtyping and facilitate the identification of specific subgroups that would benefit from tailored treatment strategies. However, the design and effectiveness of drugs based on RNA oligonucleotides are complex due to certain inherent features. These include their limited ability to penetrate cell membranes, their susceptibility to breakdown by nucleases, their low binding affinity to complementary sequences, their potential to become trapped in endosomes, their inadequate delivery to intended target tissues, and the risk of off-target effects and undesired toxicities [167]. All these obstacles mean that the use of miRNAs in therapeutic applications is still limited.

As it was mentioned, there is substantial heterogeneity in the expression of miRNAs across different BC subtypes, and even within individual tumors [168]. In addition, studies indicate that some miRNAs, which show elevated expression in cancer patients, may lack specificity [169]. This could primarily be attributed to variations in the handling of samples, the methods used for measurements, and the processes involved in data analysis [170]. Other factors such as patient-specific variations, environmental influences, and the role of the tumor microenvironment further complicate the interpretation of miRNA data [171,172]. This complexity makes it difficult to pinpoint the precise roles of specific miRNAs in the diagnostic and treatment response. Therefore, robust bioinformatics tools, such as machine and deep learning, will be essential to accurately analyze and interpret the vast amount of data generated from these studies. 

Despite the promise of these novel bioinformatics tools, there are still other technical hurdles that need to be addressed. For instance, there are technical difficulties associated with the efficient loading of miRNA into the NPs or exosomes, ensuring the stability of these vehicles in biological systems, and the control of their precise delivery to targeted cells [173,174]. Moreover, regulatory aspects related to safety, specificity, and potential off-target effects also need to be carefully considered. Thus, while the advancements are commendable, extensive further research and development are required to fully realize the potential of these delivery systems in clinical applications.

Overall, the future prospects for miRNAs in BC research and clinical applications are highly promising. Further advancements in understanding miRNAs and their mechanisms have the potential to revolutionize the management of this disease, ultimately leading to improved diagnostic approaches, personalized therapeutic interventions, and overcoming treatment resistance. Continued research efforts, the integration of multi-omics data, and the use of artificial intelligence (AI) are essential to fully unlock the potential of miRNAs in BC and translate scientific findings into clinical benefits for patients.

## 8. AI-Based Strategies for miRNA Use in BC

High-throughput technologies, such as microarray and NGS, enable the simultaneous profiling of thousands of miRNAs in BC samples. However, the presence of a multitude of miRNAs and the variations among patients create difficulties in discerning genuine associations between miRNAs and cancer from coincidental connections. AI and machine learning techniques present a promising solution for surmounting these obstacles [175,176]. By assimilating and analyzing vast datasets from diverse sources, these tools can efficiently extract meaningful features and help identify miRNA signatures, predict targets, and predict tumor-specific biomarkers [177,178,179]. Consequently, in recent times, several learning platforms and methods have been developed to bridge the gap between high-throughput miRNA data and the practical application in clinical settings [180]. For instance, machine learning methods have been used to differentiate between different stages and types of cancer based on miRNA profiles [179,181,182]. These models can be trained on a dataset where the miRNA expression profiles are known for each subtype, learning the patterns in this data, and can then be used to predict the subtype of new patients based on their miRNA expression.

AI and machine learning algorithms can also enhance the discovery process for miRNA biomarkers. Several machine learning models and algorithms, including Support Vector Machines (SVM), Hidden Markov Models, Neural Networks and Random Forest algorithms, have been effectively employed to construct predictive models [116]. By enabling early detection and risk assessment, these models assist clinicians in improving patient outcomes.

The treatment response in BC can vary widely, depending on a number of factors, including the patient’s age, stage of cancer, and tumor type. AI algorithms can integrate miRNA expression data with clinical parameters to predict treatment response and identify potential drug targets. This information can help oncologists to tailor treatment plans for improved efficacy and reduced side effects.

On the other hand, AI algorithms can also be applied to analyze miRNA–mRNA interaction networks, unraveling complex regulatory circuits involved in BC pathogenesis. In this way, AI can help us to understand the complex regulatory circuits involved in BC pathogenesis.

Despite the promising future of AI use in miRNA analysis, there are a number of challenges that need to be addressed to fully exploit the potential of these tools in BC. For example, there is still the limited availability of miRNA expression data. Currently, these datasets are often limited in size and scope, limiting the usability of AI tools. Therefore, robust miRNA expression data are needed to train AI models for cancer diagnosis, prognosis, and treatment response. In addition, the known complexity of miRNA expression regulation further complicates the integration of these tools. As mentioned above, miRNAs are regulated by a variety of factors, such as other miRNAs, proteins, and post-transcriptional modifications. This complexity may make it difficult to develop AI models capable of accurately predicting miRNA expression levels in different types of BC. On the other hand, clinical validation is essential for effectively utilizing machine learning-selected miRNA targets as biomarkers. This validation is necessary to ensure their accuracy and cost-effectiveness in fulfilling their intended roles. The successful validation will lead to improved treatment outcomes and the comprehensive management of BC, spanning from early detection to personalized therapy. Ultimately, this will enhance the quality of life for BC patients.

Despite these challenges, the use of AI and machine learning for miRNA analysis in cancer is a promising area of research. With continued advances in AI technology and the availability of larger miRNA expression datasets, AI is likely to play an increasingly important role in cancer diagnosis, prognosis, and treatment. In the near-future, advances in AI algorithms and increased collaboration between researchers, clinicians and industry stakeholders are expected to enhance the integration of AI into routine BC management. In addition, combining miRNA analysis with other omics data could provide a more complete understanding of BC biology and enable the development of targeted therapeutic approaches.

In summary, the use of AI for miRNA analysis in cancer is a promising area of research. With continued advances in AI technology and the availability of larger miRNA expression datasets, AI is likely to play an increasingly important role in cancer diagnosis, prognosis, and treatment.

## 9. Concluding Remarks

In the past few years, remarkable advancements have been achieved in identifying the source and functionalities of miRNA, opening up possibilities for their application in research and clinical settings for both healthy individuals and patients with various diseases. Particularly, circulating miRNAs show significant promise as non-invasive biomarkers for early-stage BC diagnosis, complementing the differentiation of BC based on clinical and histopathological grading. Moreover, they can help predict the probability of relapse, recurrence, and treatment responses in patients, offering valuable guidance to clinicians in devising personalized treatment strategies for individual patients. However, the use of miRNAs as potential therapeutic and/or diagnostic tools have still limitations due to the instability of miRNAs in plasma and the insecurity of viral vectors. Beyond the technological hurdles, the transition of miRNA biomarkers from the research stage to clinical application comes with its own set of challenges. These issues are primarily linked to the unreliable nature of miRNAs as distinct biomarkers, for a variety of reasons: numerous miRNAs have been identified in patients with diverse tumor types; different outcomes have been reported for the same miRNA across various studies, even for identical tumors; and distinguishing between closely related miRNAs often proves difficult. Despite these obstacles, the realm of miRNA-centered cancer treatment is quickly advancing. With ongoing research, it is anticipated that miRNAs will have a progressively significant impact on cancer detection, therapy, and prevention in the foreseeable future.

Specifically, further studies are needed to validate the promising results that have already been obtained in the use of miRNAs as non-invasive prognostic and predictive biomarkers in BC subtypes. These efforts would provide valuable complementary information on the genetic landscape and molecular characteristics of BC, thereby elucidating its underlying biological heterogeneity. Furthermore, this knowledge would play a crucial role in refining diagnostic and prognostic assessments in clinical settings, facilitating the development of personalized medicine approaches. In addition, it would be possible to optimize therapeutic options, leading to more effective and tailored treatment strategies for each patient.

To comprehensively evaluate the clinical utility of miRNAs in BC, it is essential that future studies focus on analyzing large, well-characterized and stratified patient cohorts. This approach would allow for more accurate and reliable assessments of the prognostic and predictive value of miRNAs in BC treatment. In addition, these studies should also seek to identify novel miRNAs that could serve as informative biomarkers. This exploration of unexplored miRNAs would contribute to broadening our understanding of the involvement of miRNAs in BC and potentially lead to the development of innovative diagnostic and therapeutic strategies in the future. Moreover, to thoroughly establish the clinical utility of miRNAs as diagnostic, prognostic, and predictive biomarkers, it is essential to quantify them in extensive prospective multicenter studies, encompassing diverse patient cohorts representing various tumor stages. These investigations should be conducted independently to ensure the robustness and reliability of the findings.

Overall, the use of miRNAs as biomarkers in BC is a promising area of research. Further studies are needed to validate the existing findings and to identify novel miRNAs that could be used in clinical practice. This would provide valuable insights into BC biology and could lead to improved patient outcomes.

## Figures and Tables

**Figure 1 diagnostics-13-03072-f001:**
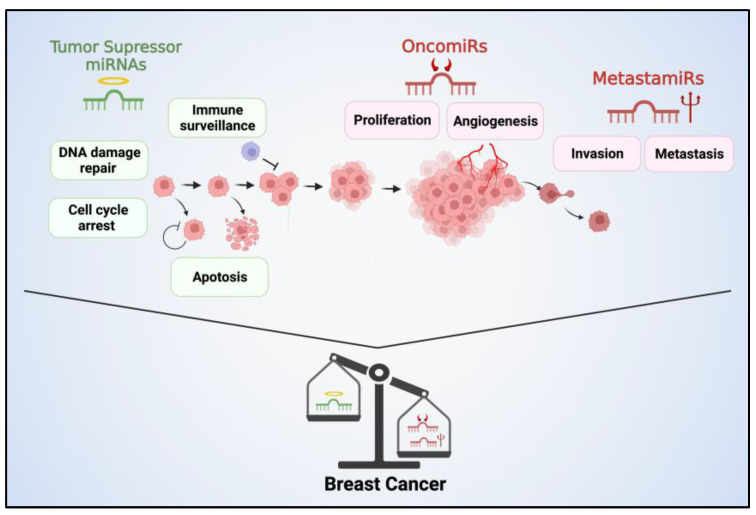
Tumor suppressor miRNAs, oncomiRs, and metastamiRs play critical roles in BC progression. Tumor suppressor miRNAs function to inhibit tumor growth and metastasis by regulating the expression of genes involved in cell cycle control, apoptosis, and DNA repair. Their reduced expression can contribute to uncontrolled cell proliferation, resistance to apoptosis, and increased genomic instability, promoting the development and progression of BC. Conversely, oncomiRs promote oncogenic signaling pathways, cell proliferation, angiogenesis, and invasion. These miRNAs are often overexpressed in BC and target tumor suppressor genes. MetastamiRs contribute to the metastatic cascade. They regulate genes involved in cell adhesion, migration, invasion, and angiogenesis, promoting the dissemination of BC cells to distant sites.

**Figure 2 diagnostics-13-03072-f002:**
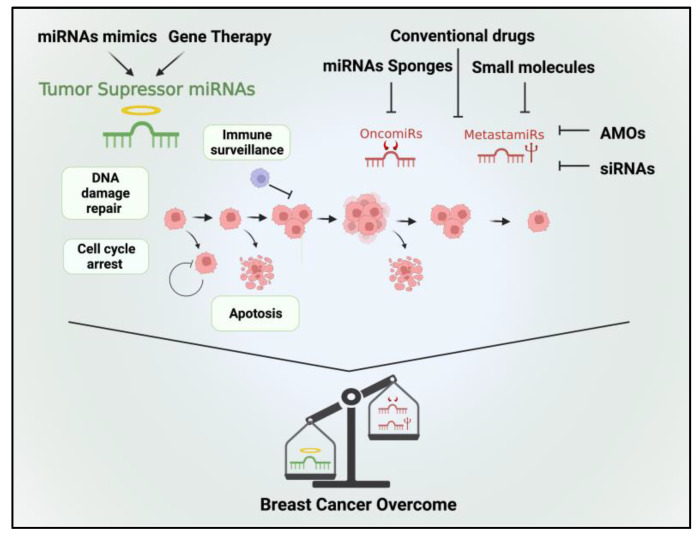
Therapeutic options targeting tumor suppressor miRNAs, oncomiRs, and metastamiRs hold promise for the treatment of BC. Tumor suppressor miRNAs can be therapeutically restored to reestablish their inhibitory functions in BC cells. Various approaches include the use of synthetic miRNA mimics or viral vectors to deliver tumor suppressor miRNAs into cancer cells. By reintroducing these miRNAs, aberrant oncogenic signaling pathways can be suppressed, leading to reduced tumor growth, enhanced apoptosis, and increased sensitivity to chemotherapy. In contrast, therapeutic strategies aiming at oncomiRs involve inhibiting their activity or expression. Thus, by inhibiting oncomiRs, cellular processes promoting tumor growth, invasion, and resistance to treatment can be attenuated. By targeting metastamiRs, it may be possible to impede key steps in the metastatic cascade, such as cell migration, invasion, and angiogenesis.

**Table 1 diagnostics-13-03072-t001:** miRNAs dysregulated in BC subtypes.

Luminal A	Luminal B	HER2+	TNBC
miR-30c-5p, miR-30b-5p, miR-182-5p, miR-200b-3p [51]		miR-520d, miR-181c, miR-302c, miR-376b, miR-30e [52]	miR-18b, miR-103, miR-107 miR-652 [53]
miR-1290 [54]		miR-4734, miR-150-5p [55]	miR-520g [52]
miR-99a, miR-125b, let-7c [56]	miR-15b-3p, miR-149-5p, miR-182-5p, miR-193b-3p, miR-200b-3p, miR-342-3p [57]	miR-125b [58]	miR-155, miR-493, miR-30e, miR-27a [59]
miR-16, miR-145, miR-155, miR451a, miR-21, miR-486 [60]	miR-342 [52]	miR-940, miR-451a, miR-16-5p and miR-17-3p [61]	miR-21, miR-221 and miR-210 [62]
miR-29c-5p, miR-130b-3p, miR-185-5p, miR-362-5p, 378a-3p [57]			miR-21-3p, miR-659-5p, miR-200b-5p [63]
miR-29a, miR-181a, miR-223, miR-652 [64]			miR-18b, miR-103, miR-107, miR-652 [53]

**Table 3 diagnostics-13-03072-t003:** Studies showing upregulated miRNAs in BC patients after recurrence.

Sample	miRNA	BC Types/Recurrence	Additional Findings	Ref.
Breast tumor samples	miR-17-5p	All/Locoregional and Metastatic	Higher miR-17-5p expression was associated with a worse 5-year RFS	[148]
FFPE breast tumor specimens/plasma	miR-30b-5p	All/Metastatic	Bone metastases and their primary tumors displayed higher miRNA expression levels	[149]
FFPE breast tumor speci-mens	miR-3651	All/local	miR-3651 may predict local control in early BC via FRMD3	[152]
Plasma	miR-21, miR-23b, miR-200c	All/NS	miR-21 and miR-200c expression was higher in patients with late relapse	[154]
Breast tumor samples	miR-9	All/Local	Expression levels of miR-9 was associated with ER status	[155]
Serum	miR-18b, miR-103, miR-107 miR-652	TNBC/Metastatic	This miRNA signature serves to stratify TNBC tumors according to their potential metastatic behavior.	[53]
Serum	miR-21-5p, miR-375, miR-205-5p, miR-194-5p	All/Locoregional and Metastatic	High levels of circulating miR-194-5p was associated with other cancer types	[156]
Serum	miR-155, miR-24	All/Metastatic	Combining the levels of miRNAs and Ki-67 expression can help to better predict the risk of relapse.	[157]
Serum	miR-122	All/Metastatic	Useful to predict metastatic recurrence in stage II-III BC patients.	[158]
FFPE primary tumor specimens	miR-4734, miR-150-5p	HER2/NS	Integrating this miRNA-based classifier into the TNM staging system could improve the ability of the TNM system to predict the prognosis of cancer patients.	[55]
Plasma	miR-221	Luminal/Local and Metastatic	miR-221 is a potential biomarker of tamoxifen resistance.	[159]
Serum and frozen tissue	miR-488-5p	All/metastatic	pre-miR-488 is an independent poor prognostic factor for RFS	[160]
Serum	miR-93-5p, miR-130a-3p, miR-17-5p, and miR-340-5p	All/NS	Authors report different miRNA expression patterns between tumors and exosomes	[161]

Abbreviations: FFPE, Formalin-fixed, paraffin-embedded. RFS, Relapse-free survival. NS, Not specified.

## Data Availability

Not applicable.

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
