# Peer review of "The Role of MicroRNAs in Breast Cancer and the Challenges of Their Clinical Application"

_diagnostics, 2023, doi:10.3390/diagnostics13193072_

Round 1

Reviewer 1 Report

In their review, the authors delve into microRNAs (miRNAs) and their impact on breast cancer. The authors highlight distinct miRNA patterns in normal and cancerous breast cells, pointing to their potential in diagnostics and therapies. They discuss miRNAs' roles as cancer promoters or suppressors and their promise as diagnostic tools. Additionally, the authors touch on the challenges of using AI for miRNA analysis in clinical applications. In total, the authors have provided a comprehensive review that effectively summarizes and discusses various miRNAs in the context of cancer. Their use of graphics to elucidate the diverse expressions and modulations of miRNAs is commendable. To enhance readability, I would suggest incorporating a tabular overview of the described miRNAs. This approach could facilitate a clearer and more organized presentation of the information contained in the review.

Author Response

Dear Reviewer.

We appreciate the time spent reviewing our manuscript. Regarding your suggestion, we have included tables to improve the clarity and organization of the information presented, providing readers with a more accessible and structured format for understanding the miRNAs discussed (See new tables 1, 2 and 3). We are confident that this addition has improved the overall quality and readability of the review.

Thank you for your valuable suggestion.

Reviewer 2 Report

Comments on Manuscript ID: diagnostics-2577742

Title: The role of microRNAs in breast cancer and the challenges of 2 their clinical application

The main problem with the manuscript is that it has almost no novelty in terms of data or conceptually in the field covered in the article

1) The authors describe in detail the biosynthesis of miRNA and how it binds to the target mRNA, but they don’t mention the fact that what determines the pairing between the miRNA and its target is a sequence of 6-8 bases called "seed”.

2) Since the authors devote a large part of the article describing the biosynthesis of microRNA and how the proteins bind to the RNA and process it, it was right to try to be precise in the figure describing the process in Figure. After solving the structure of the complex )DGCR8 with DORSHA and the pri-miRNA) The model is that two DGCR8 proteins bound to the loop structure governing the DORSHA location on the stem structure and localizing the DROSHA cutting site. These findings are described in the work of Narry Kim, one of the world's leading scientists on miRNA biosynthesis (titled: Functional Anatomy of the Human Microprocessor  Cell. 2015 Jun 4;161(6):1374-87.doi: 10.1016/j.cell.2015.05.010). work. Moreover, in 2014 she published a review on the Regulation of microRNA biogenesis (Nature Reviews Molecular Cell Biology volume 15, pages 509-524 (2014)) The authors should cite her work.

3) In lines 164-166 the author wrote the know clsifaction of “breast cancer breast cancers can be classified into different subtypes based on the expression of ER, PR, and HER2. These subtypes include ER+/PR+,  HER2-positive, and triple-negative breast cancer (TNBC)”

In the next section, they write about the miRNAs that were found to change in breast cancer and the signaling pathways in which they might or were shown to be involved. Because there are many reviews on miRNAs and miRNAs in breast cancer  I think that in order to add novelty to this review the authors should try and correlate miRNAs changes in breast cancer with specific subtypes of breast cancer. Otherwise, the novelty of this manuscript is very limited.

4) The authors review the potential therapeutic use of miRNAs in breast cancer. They pointed to a few challenges with miRNA therapeutic. All of these challenges were reviewed in many other works in the past. Phillip Sharp discusses RNAi, Nobel Prizes, and entrepreneurial science in 2005 already pointed to these delivery problems of siRNA [Drug Discovery Today

Volume 10, Issue 1, 1 January 2005, Pages 7-10].

5) The main novelty of this manuscript is the possibility of using liquid biopsies as markers for cancer stages and subtypes. However, a recent paper reviewing this was published recently [Mol Cancer. 2023 Feb 16;22(1):33. doi: 10.1186/s12943-023-01741-x. Recent advances of small extracellular vesicle biomarkers in breast cancer diagnosis and prognosis].

Author Response

"Reviewer 2:  The main problem with the manuscript is that it has almost no novelty in terms of data or conceptually in the field covered in the article"

Dear reviewer. We greatly appreciate the time and effort you dedicated to evaluate our manuscript, and we acknowledge the expertise and insights you have brought to the review process. Below you will find answers to your comments point by point.

"1) The authors describe in detail the biosynthesis of miRNA and how it binds to the target mRNA, but they don’t mention the fact that what determines the pairing between the miRNA and its target is a sequence of 6-8 bases called "seed”.

Thank you for your comment. We have taken your suggestion into account and have added the following text to lines 39-44:

"2) Since the authors devote a large part of the article describing the biosynthesis of microRNA and how the proteins bind to the RNA and process it, it was right to try to be precise in the figure describing the process in Figure. After solving the structure of the complex )DGCR8 with DORSHA and the pri-miRNA) The model is that two DGCR8 proteins bound to the loop structure governing the DORSHA location on the stem structure and localizing the DROSHA cutting site. These findings are described in the work of Narry Kim, one of the world's leading scientists on miRNA biosynthesis (titled: Functional Anatomy of the Human Microprocessor  Cell. 2015 Jun 4;161(6):1374-87.doi: 10.1016/j.cell.2015.05.010). work. Moreover, in 2014 she published a review on the Regulation of microRNA biogenesis (Nature Reviews Molecular Cell Biology volume 15, pages 509-524 (2014)) The authors should cite her work."

Thank you for your comment. We have taken your suggestion into account and have made the following changes:

I- As suggested by another reviewer, we have fused the section on the biosynthesis of miRNAs with the introduction and removed Figure 1. This change was made in the interest of improving the readability of the article.

II- The reference to the work of Narry Kim has been added to the section on the biosynthesis of miRNAs. The reference is cited in lines 57-62 (references 16 and 17). As you say, this reference is important because it provides a detailed description of the molecular mechanisms involved in miRNA biogenesis.

"3) In lines 164-166 the author wrote the know clsifaction of “breast cancer breast cancers can be classified into different subtypes based on the expression of ER, PR, and HER2. These subtypes include ER+/PR+, HER2-positive, and triple-negative breast cancer (TNBC)” In the next section, they write about the miRNAs that were found to change in breast cancer and the signaling pathways in which they might or were shown to be involved. Because there are many reviews on miRNAs and miRNAs in breast cancer  I think that in order to add novelty to this review the authors should try and correlate miRNAs changes in breast cancer with specific subtypes of breast cancer. Otherwise, the novelty of this manuscript is very limited."

We have added a new table in the manuscript describing the correlation between miRNA expression and specific subtypes of breast cancer. Please, see the new table 1.

"4) The authors review the potential therapeutic use of miRNAs in breast cancer. They pointed to a few challenges with miRNA therapeutic. All of these challenges were reviewed in many other works in the past. Phillip Sharp discusses RNAi, Nobel Prizes, and entrepreneurial science in 2005 already pointed to these delivery problems of siRNA [Drug Discovery Today Volume 10, Issue 1, 1 January 2005, Pages 7-10]."

We acknowledge your point that some of the challenges associated with miRNA therapeutics, particularly delivery issues, have been extensively reviewed in previous works, such as Phillip Sharp's 2005 article on siRNA. However, it is important to emphasize that while these challenges have indeed been addressed in the past, our manuscript offers a unique perspective by incorporating recent advancements in the field, particularly in the realm of artificial intelligence (AI). We believe this is a contribution to the current body of literature. These AI-based strategies offer a promising avenue for overcoming some of the longstanding challenges associated with miRNA therapeutics.

"5) The main novelty of this manuscript is the possibility of using liquid biopsies as markers for cancer stages and subtypes. However, a recent paper reviewing this was published recently [Mol Cancer. 2023 Feb 16;22(1):33. doi: 10.1186/s12943-023-01741-x. Recent advances of small extracellular vesicle biomarkers in breast cancer diagnosis and prognosis]."

We appreciate your reference to the recent paper, and we would like to provide further clarity regarding the approach and purpose of our review. Our manuscript is designed to offer a comprehensive and reader-friendly overview of the existing literature on the diagnostic and therapeutic applications of miRNAs in the context of breast cancer, including recent advancements. While we acknowledge that certain aspects covered in our review may have been addressed in recent publications, including the one you mentioned, it's important to emphasize that our review aims to provide a consolidated and well-contextualized perspective for our readers.

The deliberate inclusion of well-established findings and known references in our review serves the purpose of presenting a holistic view of the field. This approach ensures that readers, whether they are experts or newcomers to the subject, can gain a complete understanding of the topic. By offering this comprehensive view, we aspire to serve as a valuable resource for individuals seeking to grasp the current state of research in miRNA applications for breast cancer, even when certain aspects are well-known within the existing literature.

We sincerely appreciate the time you have taken to review our manuscript and will carefully consider whether you have any additional comments.

Reviewer 3 Report

This review focuses on the current topic of the role of microRNAs in breast cancer and the challenges of their clinical application. The review is well illustrated, but at the same time has a number of shortcomings:

- The first part of the review is extremely lengthy and describes long-known truths. Please shorten sections 1 and 2 by combining them in the introduction. The review would benefit if the authors removed Figure 1

- refresh the references. For example, reference 143 about the concentration of exosomes in the blood of breast cancer patients: the authors evaluated by indirect method in 2011, however there is a paper published in 2020 where track analysis provides more conclusive data on exosome concentration. (see DOI: 10.3390/biom10040495)

- it would be good to include summarizing tables, for example, for diagnostically important microRNAs. Such a presentation would be more concise and informative.

In general, the review makes a good impression, but requires some revision

Minor editing of English language required

Author Response

"Reviewer 3. This review focuses on the current topic of the role of microRNAs in breast cancer and the challenges of their clinical application. The review is well illustrated, but at the same time has a number of shortcomings:

"1) The first part of the review is extremely lengthy and describes long-known truths. Please shorten sections 1 and 2 by combining them in the introduction. The review would benefit if the authors removed Figure 1"

Dear Reviewer. We appreciate your constructive feedback on our manuscript. We have considered your comments and made the necessary revisions to address your concerns:

We have condensed the first and second sections of the review and integrated them into the introduction. This modification will provide a more concise and focused introduction to the topic, avoiding unnecessary repetition of long-known truths. Additionally, we have removed Figure 1 as per your recommendation. We believe this change streamlines the manuscript and enhances its readability.

"2- refresh the references. For example, reference 143 about the concentration of exosomes in the blood of breast cancer patients: the authors evaluated by indirect method in 2011, however there is a paper published in 2020 where track analysis provides more conclusive data on exosome concentration. (see DOI: 10.3390/biom10040495)."

Thank you for your suggestion. We have indeed made efforts to ensure the inclusion of the most recent and relevant research in our review. In response to your specific comment regarding reference 143 on exosome concentration in the blood , we have updated it to include the paper with the DOI: 10.3390/biom10040495, which provides more conclusive data on exosome concentration based on track analysis (See new reference 120).

While we have refreshed many references to incorporate the latest findings and advancements in the field, we have also retained certain older references that we consider iconic and foundational to the subject matter. These seminal works have played a pivotal role in shaping the field and remain important points of reference for readers seeking a comprehensive understanding of the topic.

"3- it would be good to include summarizing tables, for example, for diagnostically important microRNAs. Such a presentation would be more concise and informative."

Thank you for your suggestion regarding the inclusion of summarizing tables for diagnostically important microRNAs in our manuscript. We have taken your feedback into account and have created three new tables to provide a concise and informative presentation of these microRNAs. These tables offer a clear overview of diagnostically significant miRNAs, enhancing the accessibility and readability of the manuscript. We believe that this addition will greatly benefit our readers in understanding the key microRNAs in the context of our review.

We appreciate your valuable input, and we hope that these new tables enhance the quality and comprehensibility of our manuscript. If you have any further comments or recommendations, please feel free to share them with us.

Round 2

Reviewer 2 Report

The authors answer my comment and therefore I do not have any new comments.

Still, my main problem with this manuscript is the degree of novelty, which is quite limited.